# Efficiency of the Moscow Stock Exchange before 2022

**DOI:** 10.3390/e24091184

**Published:** 2022-08-25

**Authors:** Andrey Shternshis, Piero Mazzarisi, Stefano Marmi

**Affiliations:** Quantitative Finance Research Group, Scuola Normale Superiore, Piazza dei Cavalieri 7, 56126 Pisa, Italy

**Keywords:** Shannon entropy, market efficiency, volatility estimation, price staleness, stock market clustering, Kullback–Leibler divergence

## Abstract

This paper investigates the degree of efficiency for the Moscow Stock Exchange. A market is called efficient if prices of its assets fully reflect all available information. We show that the degree of market efficiency is significantly low for most of the months from 2012 to 2021. We calculate the degree of market efficiency by (i) filtering out regularities in financial data and (ii) computing the Shannon entropy of the filtered return time series. We developed a simple method for estimating volatility and price staleness in empirical data in order to filter out such regularity patterns from return time series. The resulting financial time series of stock returns are then clustered into different groups according to some entropy measures. In particular, we use the Kullback–Leibler distance and a novel entropy metric capturing the co-movements between pairs of stocks. By using Monte Carlo simulations, we are then able to identify the time periods of market inefficiency for a group of 18 stocks. The inefficiency of the Moscow Stock Exchange that we have detected is a signal of the possibility of devising profitable strategies, net of transaction costs. The deviation from the efficient behavior for a stock strongly depends on the industrial sector that it belongs to.

## 1. Introduction

When prices reflect all available information, the market is called efficient [1]. One way to claim the efficiency of a market is by testing the Efficient Market Hypothesis (EMH). In its weak form, the EMH considers that the last price incorporates all the past information about market prices [2]. If the weak form of EMH is rejected, previous prices help to predict future prices. For traders, market efficiency means that analyzing the history of previous prices does not help to design a strategy that produces abnormal profits. For a company issuing shares, market efficiency means that the cost of its share already reflects all information about the valuation and decisions of the company. The EMH is of great interest also in research. Mathematical models of an asset price are usually based on the assumption that the price follows a martingale: the expected value of a future price is the current value of the price. If the EMH is rejected, there should be an estimation of the future price that is better than its current value. In such a case, new models should be created.

A review of studies confirming the EMH was presented by Fama in 1970 [2] and then in 1991 [3]. The martingale hypothesis was also tested later. It was shown that the efficiency of a market depends on the development of the country [4]. Moreover, the martingale hypothesis was confirmed on short time intervals, but it may be violated on longer intervals [5]. In addition, there is a range of strategies designed to increase an expected profit. High-frequency and algorithmic trading strategies are discussed in [6]. Statistical and machine learning methods for high frequency trading are reviewed in [7]. The existence of such profitable strategies contradicts the Efficient Market Hypothesis. According to Grossman and Stiglitz [8], the degree of market inefficiency determines the effort investors are willing to expend to gather and trade on information.

The goal of this paper is to investigate the degree of stock market efficiency of the Moscow stock Exchange using the Shannon entropy. We quantify the degree of market inefficiency and the degree of price randomness. We aim to distinguish between price predictability due to stylized facts of financial time series [9] and due to market inefficiency. In particular, we consider volatility clustering and price staleness as data regularities needed to be filtered out. Based on the behavior of stock prices, we group them into clusters using several measures. Combining stocks into one cluster means a common price behavior that moves prices away from complete randomness.

A range of methods is used to measure a degree of market efficiency. In particular, Cajueiro and Tabak used the Hurst exponent and R/S statistics to rank efficiency of markets [10,11]. The Hurst exponent was measured on Bitcoin data to compare it with mature markets [12]. A generalized version of the use of the Hurst exponent, multifractal detrended fluctuation analysis, was applied to investigate the efficiency of stock and credit markets [13]. The algorithmic complexity of return time series was applied to measure the efficiency of financial markets [14] and to check the Efficient Market Hypothesis [15]. Finally, the Shannon entropy as a measure of randomness is used in a range of articles; see [16,17,18]. The general idea of these methods is to compare the characteristic of the time series with the value corresponding to a completely random process. In our study, we use Monte Carlo simulations to determine which deviations from a completely random process are statistically significant.

Before estimating the degree of market efficiency, we need to dispose of regularities that make prices more predictable but that do not imply any profitable strategies. A method for filtering regularities was introduced in [19]. However, such a process of filtering has not usually been applied in other research studies. In fact, deviations of price behavior from perfect randomness may be the result of some known regularity pattern, such as volatility clustering or daily seasonality, but not a signal of market inefficiency. One of the innovations of this article is a new method for filtering data regularities, allowing the estimation of volatility and a degree of price staleness minute by minute.

We process data by filtering regularities of financial time series including volatility clustering and price staleness. Price staleness is defined as a lack of price adjustments yielding 0-returns. Traders may trade less because of high transaction costs and so the price does not update. See [20] for more details. Price staleness produces an extra amount of 0-returns called *excess 0-returns*. The other source of 0-returns in the time series is price rounding. Estimations of volatility and degree of price staleness are mutually connected: Excess 0-returns appear due to price staleness tend to underestimate volatility. At the same time, volatility estimation is needed to calculate the expected amount of 0-returns due to rounding.

One method for estimating volatility in the presence of excess 0-returns was presented in [21]. It uses expectation-maximization algorithm [22] to estimate returns in the places of all 0-returns and uses the GARCH(1,1) model to estimate volatility [23]. The maximization of the likelihood function appearing at each step of the considered algorithm requires several parameters for numerical optimization. If the estimation of volatility is sensitive to these parameters, which are user-defined, then they may affect the entropy of returns standardized by volatility and the amount of 0-returns in the time series. In this article, we suggest a modification of moving average volatility estimations that require an adjustment of the only parameter that can be defined using out-of-sample testing. The idea is to adopt a simple method for volatility estimation such that price staleness is taken into consideration. Moreover, while estimating volatility, we filter out excess 0-returns.

The degree of market efficiency has been measured for many countries. Stock indices for 20 countries were considered in [24]. The efficiency of 11 emerging markets and the US and Japan markets was measured in [10]. US stock markets were considered in a recent paper [25]. A review of articles about Baltic countries was presented in [26]. A degree of uncertainty of Chinese [27], Tunisian [28], Mexican [29], and Portuguese [30] stock markets was also considered by using entropy measures. However, the efficiency of the Russian stock market has not yet been analyzed. In this paper, we present an analysis of market efficiency based on the estimation of Shannon entropy for a group of 18 stocks of Russian companies from five industries.

Our paper introduces four original contributions in the field. First, we construct a method for filtering out heteroskedasticity and price staleness. This filtering process helps identify a true degree of market inefficiency. Second, we calculate the degree of market inefficiency for the previous decade using monthly intervals. We conclude that the degree of market inefficiency for the Moscow Stock Exchange was greater than 80%. Third, we determine which pair of stocks exhibits the largest amount of inefficiency, as measured by estimating Shannon’s entropy on their high-frequency price time series. We show that months where the predictability of stock prices attains its maximum cluster together. We find out the form of behavior that is repeated most often with respect to stocks for inefficient time periods. Finally, we estimate the closeness of price movements using two measures of entropy. Based on these results, we cluster together groups of stocks for which the efficient market hypothesis is rejected, thus pointing out how market inefficiency displays some dependence on the financial sector they belong.

The article is organized as follows. Section 2 describes the dataset and the method for filtering data regularities and calculating the Shannon entropy. Section 3 presents the results on simulated and real data. Section 4 concludes the paper.

## 2. Materials and Methods

Our main goal is to measure a degree of efficiency of the Moscow Stock Exchange. The data taken for the study are reviewed in the next section. All data processing and computing can be divided into three stages. First, we filter data regularities from financial time series. Then, we calculate the degree of efficiency of the market using the Shannon entropy. Finally, we use the resulting time series to cluster stocks using Kullback–Leibler distance discussed in Section 2.4.

### 2.1. Dataset

We study the Moscow Stock Exchange. We consider close prices aggregated at one-minute time scale. In particular, we select only minutes of the main trading session from 10:00 to 18:40. The time interval covers ten years from 2012 to 2021. The time period is divided into monthly time intervals. We take 18 companies, 16 of them are from five sectors: oil industry, metallurgy, banks, telecommunications, and electricity. All stocks are listed in Table 1 (there are 2520 trading days. Assuming that there are 520 min in each trading day, there are 1310400 trading minutes in total. We use the Brownlees and Gallo’s algorithm of an outlier detection [31]. See details in Section A.1). All data are provided by Finam Holdings (https://www.finam.ru/, accessed on 17 August 2022).

### 2.2. Apparent Inefficiencies

To estimate a degree of market efficiency, we first should eliminate the known patterns of predictability, such as a daily seasonality. Financial agents operating in the market tend to trade less in the middle of a day. It is reflected in prices, but again, this pattern in trading volume should be filtered out to detect genuine patterns of inefficiency. Other known regularities include volatility clustering, price staleness, and microstructure noise. See Appendix A for a guide on filtering out apparent inefficiencies. The contribution of this article is that it devises a simple method for filtering volatility clustering and price staleness. One of the methods used to estimate volatility is the exponentially weighted moving average (EWMA). It is described in the next section.

#### 2.2.1. EWMA

We define price returns as $r_{t}=\ln\left(\frac{P_{t}}{P_{t-1}}\right)$, where $P_{t}$ is the last price available at time *t*, and ln() is the natural logarithm. In order to estimate volatility $\sigma_{n}$, we apply the exponentially weighted moving average [32] of values $\mu_{1}^{-1}\left|r_{i}\right|$, i<n, where $\mu_{1}=\sqrt{\frac{2}{\pi }}$.
(1)$${\bar{\sigma }}_{n}=Sig_{1}\left(\alpha ,r_{n-1},{\bar{\sigma }}_{n-1}\right)=\alpha \mu_{1}^{-1}\left|r_{n-1}\right|+\left(1-\alpha \right){\bar{\sigma }}_{n-1}$$

This form of exponential moving average was used in [19]. Here, $E[|r_{n}\left|\right]=\mu_{1}\sigma_{n}$ is used assuming that returns are normally distributed, $r_{n}∼N\left(0,\sigma_{n}\right)$. More weights are provided for the more recent data. An alternative formula based on expectation $E\left[r_{n}^{2}\right]=\sigma_{n}^{2}$ is described as follows.
(2)$${\bar{\sigma }}_{n}^{2}=Sig_{2}\left(\alpha ,r_{n-1},{\bar{\sigma }}_{n-1}\right)=\alpha r_{n-1}^{2}+\left(1-\alpha \right){\bar{\sigma }}_{n-1}^{2}.$$

A large value of return increases the value of volatility. The current value of volatility reflects all available values of returns and changes slowly if the value of α is small.

We follow the approach suggested by [33] (p. 97) to find optimal values of α in Equations (Equation 1) and (Equation 2). The value of α is selected so that it minimizes error $Er_{\sigma }\left(\alpha \right)=\sum_{i}{\left({\bar{\sigma }}_{i}^{2}-r_{i}^{2}\right)}^{2}$. In order to minimize $Er_{\sigma }\left(\alpha \right)$ as a function of the only parameter 0<α<1, we apply Brent’s algorithm [34] (the method is available in Python by using the function scipy.optimize.minimize_scalar. Alternatively, we could use the golden-section search [35] that requires the boundary of the search and the only parameter for the stopping criteria). We modify the exponential moving average method in Section 2.2.3 so that it removes a bias due to the effect of price staleness discussed in the next section.

#### 2.2.2. Estimation of Price Staleness

Let us define an efficient price, $P^{e}$, as a continuous process following a Geometric Brownian Motion.
$$P_{t}^{e}=P_{0}^{e}+\int_{0}^{t}\sigma_{s}P_{s}^{e}dW_{s}$$

An observed price moves along a discrete grid. Possible price values are multiples of the tick size, *d*.
$$P_{t}=d\cdot [\frac{P_{t}^{e}}{d}]$$

If the efficient price changes insignificantly, the return of the rounded price will be equal to 0. Analogically, if the return of rounded price is 0, the return of efficient price has a value close to 0. We use Equation (Equation 3) to estimate the probability that a return of rounded price has zero value:(3)$$p_{i}=erf\left(R_{i-1}\right)+\frac{1}{R_{i-1}\sqrt{\pi }}\left(\exp\left(-R_{i-1}^{2}\right)-1\right),$$
where $R_{i}=\frac{d}{{\bar{P}}_{i}{\bar{\sigma }}_{i}\sqrt{2\Delta }}$ and erf(x) are Gaussian error functions; *d* is a tick size (we estimate the tick size using a two-step procedure for each month. First, we find the amount of significant digits in price. Then, we determine the most frequent increment in ordered prices); Δ is a time step (the time step between the end and start of the main trading session is set as 1 min. Moreover, we consider any time gap without trading more than 2 h as the closure of the market. We set the time step to be equal to 1 min for these gaps); $\bar{P}$ is a rounded price; $\bar{\sigma }$ is an estimation of volatility [36]. It is obtained by considering the probability that a price following a Geometric Brownian Motion moves less than one tick size, assuming that price increments are normally distributed.

There is another source for obtaining 0-returns, namely price staleness. Price staleness represents a regularity pattern of the dynamics; namely, the fundamental (efficient) price of an asset is not updated because of a number of reasons, such as no transactions because of high cost, which makes trading unprofitable for agents. See [20] for more details. This results in a persistence pattern of (“*excess*”) 0-returns. Such a pattern, for example, tends to reduce any estimation of the volatility. Therefore, we need to filter out 0-returns due to price staleness while retaining 0-returns due to rounding for a genuine estimation of volatility.

We save 0-returns in the amount of the sum of past values of the probability in Equation (Equation 3) [36]. We set other 0-returns as missing values. We adopt this method to estimate the degree of price staleness together with volatility in the next section.

#### 2.2.3. Modification of EWMA

In this Section, we present a modification of the EWMA that takes into consideration the effect of price staleness. Our modification of the EWMA is based on the suggestion for estimating volatility $\sigma_{n}$ as ${\bar{\sigma }}_{n-1}$ (i.e., by setting α=0), if the value of $r_{n-1}$ is missing because of price staleness. That is, there is no new information from returns to update the value of volatility.

Initially, the expected amount of 0-returns due to rounding is $N_{save}=0$. Thus, each appearance of 0-returns does not affect the value of volatility. A 0-return is defined as a value due to rounding and is saved in the sequence if the sum of all $p_{i}$ (Equation (Equation 3)) moves to a new integer value. Other details and the algorithm of volatility estimation can be found in Appendix B.

We update the estimation of volatility and price staleness minute-by-minute. This method has the clear advantage of making the online inference possible by processing data in real time.

### 2.3. Calculating a Degree of Market Inefficiency

#### 2.3.1. The Shannon Entropy

A degree of randomness of price returns is assessed by Shannon entropy. The entropy of a source is an average measure of the randomness of its outputs [37].

**Definition** **1.**
*Let $X=\{X_{1},X_{2},.\cdots \}$ be a stationary random process with a finite alphabet A and a measure μ. An n-th order entropy of X is*

$$H_{n}\left(\mu \right)=-\sum_{x_{1}^{n}\in A^{n}}\mu \left(x_{1}^{n}\right)\log\mu (x_{1}^{n})$$

*with the convention 0log0=0. The process entropy (entropy rate) of X is*

$$h\left(\mu \right)=\lim_{n\to \infty }\frac{H_{n}\left(\mu \right)}{n}.$$



#### 2.3.2. Discretization

The Shannon entropy is computed over a finite alphabet. To measure Shannon’s entropy, we need to keep the length of blocks of symbols, *k*, sufficiently large. The predictable behavior of returns can be seen on blocks of greater length and may not be noticeable on blocks of smaller length. For this reason, we consider 3-symbol and 4-symbol discretizations using empirical quantiles:$$\begin{array}{c} s_{t}^{\left(3\right)}=\left\{\begin{array}{c} 1,r_{t}\leq \theta_{1},\\ 0,\theta_{1}<r_{t}\leq \theta_{2},\\ 2,\theta_{2}<r_{t}, \end{array}\right. \end{array}\;\begin{array}{c} s_{t}^{\left(4\right)}=\left\{\begin{array}{c} 0,r_{t}\leq Q_{1},\\ 1,Q_{1}<r_{t}\leq Q_{2},\\ 2,Q_{2}<r_{t}\leq Q_{3},\\ 3,Q_{3}<r_{t}, \end{array}\right. \end{array}$$
where $\theta_{1}$ and $\theta_{2}$ are tertiles and $Q_{1}$, $Q_{2}$, and $Q_{3}$ are quartiles. The tertiles divide data into three equal parts. The quartiles divide data into four equal parts. $Q_{2}$ is also the median of the empirical distribution of returns. For the later analysis, we will need a discretization describing the behavior of a pair of stocks:(4)$$\begin{array}{c} s_{t}^{\left(p\right)}=\left\{\begin{array}{c} 0,r_{t}^{\left(1\right)}\leq m_{1}\;\mathrm{and}\;r_{t}^{\left(2\right)}\leq m_{2},\\ 1,r_{t}^{\left(1\right)}\leq m_{1}\;\mathrm{and}\;r_{t}^{\left(2\right)}>m_{2},\\ 2,r_{t}^{\left(1\right)}>m_{1}\;\mathrm{and}\;r_{t}^{\left(2\right)}\leq m_{2},\\ 3,r_{t}^{\left(1\right)}>m_{1}\;\mathrm{and}\;r_{t}^{\left(2\right)}>m_{2}, \end{array}\right. \end{array}$$
where $r_{t}^{\left(1\right)}$ and $r_{t}^{\left(2\right)}$ are two time series of price returns, and $m_{1}$ and $m_{2}$ are their medians.

#### 2.3.3. The Estimation Of Entropy

Let $x_{1}^{n}\in A^{n}$ be the sequence of length *n* generated by an ergodic source μ from the finite alphabet *A*, where $x_{i}^{i+k-1}=x_{i}\ldots x_{i+k-1}$. There are possible missing values in the sequence generated independently from $x_{1}^{n}$. We consider all blocks of length *k* that do not contain missing values. We take the following:(5)$$k=max(K:K<\left\lfloor \log\left(n_{b}\left(K\right)\right)\right\rfloor ),$$
where $n_{b}\left(k\right)$ is the number of blocks of length *k*. The restriction on a value of *k* allows having enough blocks to estimate probabilities appearing in *k*-th order entropy [38]. The base of the logarithm is the size of alphabet *A* (3 or 4).

For each $a_{1}^{k}\in A^{k}$, empirical frequencies are defined as follows.
$$f\left(a_{1}^{k}|x_{1}^{n}\right)=\#\left\{i\in \left[1,n-k+1\right]:x_{i}^{i+k-1}=a_{1}^{k}\right\}.$$

Empirical frequencies are the actual amount of each block from $A^{k}$ in the data. By considering an empirical k-block distribution as
(6)$${\hat{\mu }}_{k}\left(a_{1}^{k}|x_{1}^{n}\right)=\frac{f(a_{1}^{k}|x_{1}^{n})}{n_{b}},$$
an empirical *k*-entropy is defined by the following.
$${\hat{H}}_{k}\left(x_{1}^{n}\right)=-\sum_{a_{1}^{k}}{\hat{\mu }}_{k}\left(a_{1}^{k}|x_{1}^{n}\right)\log\left({\hat{\mu }}_{k}\left(a_{1}^{k}|x_{1}^{n}\right)\right)=\log\left(n_{b}\right)-\frac{1}{n_{b}}\sum_{i=1}^{M}f_{i}\log f_{i}.$$

The estimation of the process entropy is described as follows.
$${\hat{h}}_{k}=\frac{{\hat{H}}_{k}}{k}.$$

See [38] for the proof of the consistency of this estimator and [36] for the case of missing values. Since the sequence is finite, the estimation of entropy is underestimated. To remove this bias, we use the correction for the entropy estimation introduced in [39,40]:(7)$${\hat{H}}_{k}^{G}=\log\left(n_{b}\right)-\frac{1}{n_{b}}\sum_{i=1}^{M}f_{i}\log\left(\exp G(f_{i})\right),$$
where the sequence G(i) is defined recursively as
$$\begin{array}{cc} G(1) & =-\gamma -\ln(2)\\ G(2) & =2-\gamma -\ln(2)\\ G(2n+1) & =G(2n)\\ G(2n+2) & =G\left(2n\right)+\frac{2}{2n+1},\;n\geq 1 \end{array}$$
with the Euler’s constant γ=0.577215….

#### 2.3.4. Detection of Inefficiency

We need to perform three steps to determine if the time interval is efficient or not. First, we filter out apparent inefficiencies (see Appendix A). Then, we estimate the entropy of the filtered return time series using Equation (Equation 7). Finally, we determine if the value of entropy is significantly lower relative to the case of perfect randomness. We detect inefficiency in the time interval using Monte Carlo simulations. We regard a Brownian motion as absolutely unpredictable. First, we define the length of sequences as $l=n_{b}\left(k\right)+k-1$. Then, we simulate $10^{4}$ realizations of Brownian motions with Gaussian increments and the length *l*. For each realization, we calculate entropy using 3- and 4-symbol discretizations. Then, we find the first percentile of the obtained entropies for each discretization. These percentiles are the bounds of 99% of the Confidence Interval (CI) for testing market efficiency. Finally, we define an *efficiency rate* as the ratio of the entropy of the time interval and the bound of CI. If the efficiency rate is less than 1 for at least one type of discretization; *we define the time interval as inefficient*. We provide testing for inefficiency twice using different discretizations because the unique testing may not be robust. See an example in Appendix C.

### 2.4. Kullback–Leibler Distance

In addition to estimating the entropy of one time series, we can also consider the difference between two time series. Kullback–Leibler divergence [41] is used to measure similarity between two distributions for two discrete probability distributions *P* and *Q*.
$$KL\left(P|Q\right)=\sum_{i}p_{i}\log\frac{p_{i}}{q_{i}}$$

We use $p_{i}$ and $q_{i}$ as empirical probabilities obtained in Equation (Equation 6). Since the Kullback–Leibler divergence is asymmetric, we consider the distance between two time series proposed in [42].
(8)$$D\left(P,Q\right)=\frac{KL(P|Q)}{H^{G}\left(P\right)}+\frac{KL(Q|P)}{H^{G}\left(Q\right)}$$

The greater the distance of D(P,Q), the more probability distributions *P* and *Q* differ.

## 3. Results

### 3.1. Simulations

The aim of this section is to assess the accuracy of the estimation of volatility and the degree of price staleness. We will choose the method that produces the least amounts of error with respect to the estimation for further analysis on real data. We take the following model of an observed price ${\tilde{P}}_{t}$ and t=1…2N:$$\begin{array}{cc} P_{t} & =\int_{0}^{t}\sigma_{s}P_{s}dW_{s}^{1}\\ {\tilde{P}}_{i} & =P_{i}\left(1-B_{i}\right)+{\tilde{P}}_{i-1}B_{i}\\ q_{t} & =q_{0}+\int_{0}^{t}\mu_{s}ds+\int_{0}^{t}\nu dW_{s}^{2} \end{array}\;\begin{array}{c} B_{i}=\left\{\begin{array}{c} 1\;\mathrm{with}\;\mathrm{probability}\;q_{i}\\ 0\;\mathrm{with}\;\mathrm{probability}\;1-q_{i} \end{array}\right. \end{array}$$
where $W^{1}$ and $W^{2}$ are two independent Brownian motions with a length of 2N, $N=10^{5}$; a price of $P_{0}=100$; and $\nu =10^{-4}$. B=1 stands for the case when price is not updated due to price staleness (see [20,43]). Prices are rounded to two digits; thus, the tick size is d=0.01. We consider four choices for $q_{t}$ and $\sigma_{t}$ listed below.
$$\begin{array}{cc} q_{t}^{1} & =0\\ q_{t}^{2} & =0.1+\int_{0}^{t}\nu dW_{s}^{2}\\ q_{t}^{3} & =0.2+\int_{0}^{t}\nu dW_{s}^{2}\\ q_{t}^{4} & =0.2+\int_{0}^{t}\mu_{s}^{4}ds+\int_{0}^{t}\nu dW_{s}^{2}\\ \mu_{t}^{4} & =0.8\pi /N\cos(8t\pi /N)\\ \sigma_{t}^{1} & =5\times 10^{-4}\\ \sigma_{t}^{2} & ∼ARCH(1.75\times 10^{-7},0.2,0.1)\\ \sigma_{t}^{3} & ∼GARCH(1.25\times 10^{-8},0.1,0.85)\\ \sigma_{t}^{4} & ∼GARCH(1.25\times 10^{-8},0.15,0.8) \end{array}$$

For price staleness, we consider four cases: the absence of price staleness; two stochastic probabilities with different constant means; a periodic mean. For all four cases of volatility, the unconditional expected value of $\sigma_{t}$ is $5\times 10^{-4}$. The first choice of volatility is a constant. Then, we consider the ARCH model [44] with two lagged values, where 0.2 and 0.1 correspond to the first and the second lags, respectively. Volatility values directly depend only on the previous returns values. The dependency on the previous return should be reflected in the value of smoothing parameter. The third and fourth choices are GARCH(1,1) models [23], where the last parameter (0.85 or 0.8) stands for the coefficients for lagged variances. We consider two sets of parameters for a GARCH model, giving less persistence to the fourth model.

We divide the data into two equal parts with the size *N*. The first part is a training set for finding optimal values of α from Equations (Equation 1) and (Equation 2). The second part is a testing set for calculating errors represented in Table 2 and Table 3. We compare two methods that use $Sig_{1}$ and $Sig_{2}$ for volatility estimation. For each method, we find the optimal value of α. In addition, we set a fixed value of alpha, α=0.05, as a benchmark for the comparison. We also apply non-modified EMWA estimation from Section 2.2.1 with selected optimal value of α to show the contribution of 0-filtering to the accuracy of volatility estimation. We simulate $10^{3}$ prices for each model.

Table 2 represents a mean absolute percentage error (MAPE) that is $\frac{1}{N}\sum_{i}\left|\frac{{\bar{\sigma }}_{i}-\sigma_{i}}{\sigma_{i}}\right|$ for six different approaches. These approaches differ in the choice of a function for volatility, the value of α, and the presence of missing values. Table 3 represents three values for each of the two methods using $Sig_{1}$ and $Sig_{2}$ for volatility estimation. The first value is the optimal value of α. The second is $Er_{N}=\left|\frac{N_{round}N_{A}}{N_{0}N}-1\right|$, where $N_{A}$ is the amount of remaining non-missing returns, $N_{round}$ is the amount 0-returns that would appear due to rounding (before adding the effect of staleness in the simulated data), $N_{A}$ is the amount of non-missing returns, and $N_{0}$ is the amount of 0-returns. $Er_{N}$ represents the absolute error of the proportion of 0-returns that remain in the data and are defined as 0-returns due to rounding. The third value is the proportion of data set as missing values (that is, $1-\frac{N_{A}}{N}$).

It can be seen from Table 2 that the method that more often produces the lowest value of MAPE is with fixed α=0.05 and $Sig_{1}$ used for volatility estimation. Moreover, for almost all cases, 0-filtering makes the volatility estimate more accurate. The error of the amount of 0-returns due to rounding is smaller for the function $Sig_{1}$ than for the function $Sig_{2}$ for all 16 cases.

After the comparison of the two functions of volatility estimation, we decide to use $Sig_{1}$, which uses absolute values of returns, in the next sections. For the rest of the paper, we fix the value of α as 0.05 for the simplicity of further analysis.

### 3.2. Moscow Stock Exchange

We calculate 18·120=2160 efficiency rates for each type of discretization, where 18 is the amount of stocks and 120 is the amount of months in 10 years. We define a degree of inefficiency as the fraction of 2160 months that are defined as inefficient according to Section 2.3.4. The degree of inefficiency for the chosen group of stocks traded at Moscow Exchange is 0.823. In our previous study, ref. [36] we found that the degree of inefficiency for the U.S. ETF market is about 0.11 for monthly time intervals and the 3-symbol discretization only. This difference in the degrees of inefficiency can be explained by the hypothesis that developed markets have a high level of efficiency. W. A. Risso reached this conclusion in the article [24]. The degree of inefficiency for each stock and discretization is presented in Table 4. We notice that 4-symbol discretizations contribute to a larger amount of inefficient months compared to the 3-symbol discretization. That is, the 4-symbol discretization appears to have a more predictable structure than 3-symbol discretizations.

Figure 1 shows the minimum value of efficiency rates among all months for each stock.

There are two most notable deviations from one for MLTR stocks (Mechel, mining and metals company) and RSTI (Rosseti, power company). We investigate them in the next section. For the other 16 stocks, the minimum value of efficiency rate is attained for the AFLT stock, and it is equal to 0.933 (0.964) for three (four) symbols.

#### Analysis of MLTR and RSTI

We plot the values of efficiency rates for monthly intervals for the MLTR and RSTI stocks. See Figure 2 and Figure 3.

Both types of discretization show coherent results. For MLTR, there are two notable decreases in the efficiency rates at the beginning of 2014 and in the middle of 2016. For both types of discretizations, the eight months with the lowest efficiency rate (in the ascending order of time) include January–February and May–October of 2014. For each month, we write down the most frequent block of symbols in Table 5. Note that block 1111 for the 4-symbol discretization appears as the most frequent for 6 months out of 8 for MLTR. The block denotes a slight decrease in price for 4 min in a row. The meaning of the last two columns is discussed later.

For RSTI, there are two sharp decreases in 2014 and 2015. There are 11 months that have the lowest efficiency rates that are in common for both discretizations. These months are April–September of 2014 and June–October of 2015. Note that these inefficient months cluster together and are not distributed uniformly among the entire time period of 10 years. This is the signal of a market condition that affects the inefficiency of the stocks for more than one month.

We construct a simple trading strategy on discretized returns to test the predictability of future returns. We consider blocks of length 4 obtained by the 4-symbol discretization. For each month, we divide blocks into two halves. The discretization is made using only the first half of a month. We consider the sequences of the first three symbols of each block. If the empirical probability of obtaining 0 or 1 after the sequence of three symbols in the first half is greater than 0.5, this sequence is from group D (decreasing). If the empirical probability of obtaining 2 or 3 after the sequence of three symbols is greater than 0.5, this sequence is from group I (increasing). Then, for the second half of the month, we determine a success if symbols 0 or 1 follow a sequence from group D or if symbols 2 or 3 follow a sequence from group I. Then, we calculate the fraction of successes. Thus, it is the probability of making a profit: sell after group D or buy after group I. In the case of market efficiency, this probability is equal to 0.5. For example, we expect that after 111, the next symbol would be 1 according to Table 5. That is, after this block, a trader can sell a stock. The fourth column of the Table 5 shows the results for a filtered return time series. The fifth column stands for the original return time series.

For all cases, the probability is greater than 0.5. Obviously, the probabilities for the original return time series are greater than those for the filtered return time series. The reason is that predictability for the original return time series follows from the sources of apparent inefficiencies.

The same analysis is performed for the RSTI stock. Eleven months with the lowest efficiency rates are presented in Table 6. For the RSTI stock, the simple trading strategy provides the fraction of successes (of predicting increases and decreases in price) greater than 0.5 for all 11 months. The frequent behavior of the price of RSTI during the chosen months is a slight increase in price for several minutes in a row denoted by symbol 2.

The simple trading strategy is an illustrative example of market inefficiency. In fact, such a strategy could result in no profit when used in practice because it does not take into account the costs of transaction and other trading frictions. Moreover, the filtering of daily seasonality pattern is made by using the entire period of analysis. That is, this method cannot be applied in real time. Finally, we consider blocks containing only observed returns by neglecting the missing values from the analysis. Thus, the application of such a strategy in practice should be integrated with the case when a missing value follows a sequence of three symbols.

### 3.3. Stock Market Clustering

Most of the month-long time intervals are identified as inefficient. However, is there some dependence between two stocks that are inefficient at the same time?

#### 3.3.1. Kullback–Leibler Distance

We measure the similarity of discretized filtered returns by using the Kullback–Leibler (KL) distance (Equation (Equation 8)). We use *k*, the length of blocks, as the maximum value suitable for both sequences according to Equation (Equation 5). The 4-symbol discretization is used. The Kullback–Leibler divergence DL(P|Q) is calculated using empirical frequencies. The entropy rates are calculated using Equation (Equation 7). Using the Kullback–Leibler distance for all pairs of stocks, we cluster them in three groups using hierarchical clustering with the UPGMA algorithm [45]. This algorithm is implemented by using the Python function cluster.hierarchy.dendrogram with the argument distance=average. The result is in Figure 4. Combining companies into one cluster means that their stocks have a common behavior that is not related to the value of volatility, the degree of price staleness, and the structure of microstructure noise.

It can be seen that banks and oil companies are clustered together (right). There is a group of four stocks (RTKM, HYDR, AFLT, and MGNT) that have nothing in common at first glance. The remaining group (left) mainly consists of metallurgy companies. However, there is no visible distinction between the stocks of banks and oil companies. According to the clustering tree, two telecommunications companies differ significantly, as well as electricity companies.

Finally, two stocks with the lowest efficiency rates, RSTI and MLTR, are the furthest (in the sense of KL distance) from any other stock. That is, there are no stocks that behave similarly to these two stocks.

#### 3.3.2. Entropy of Co-Movement

Now, we consider another measure of difference between two stocks: the entropy of co-movement. We calculate the Shannon entropy of the discretization describing the movement of a pair of prices presented in Equation (Equation 4). We consider only minutes that are in common for both stocks. For these minutes, we consider values of residuals obtained after ARMA fitting. The result is in Figure 5.

Two companies related to telecommunications are a separate cluster. Three metallurgy companies (MAGN, CHMF, and NLMK) also cluster together. Stocks relating to oil and bank companies form the other cluster. The same cluster, with the exception of the TATN (oil industry), was also formed in the previous section. The “closeness” of stocks GAZP and SBER is detected either in this and in the previous section. The three stocks on the left that join other stock clusters last are the stocks with the lowest efficiency rates.

Some clusters may form on the basis that companies belong to the same industry. The division of companies into industries is noticeable from the dendrogram in Figure 5. However, this criterion does not explain all clusters. For instance, GMKN from metallurgy is in the cluster of oil companies and banks.

## 4. Discussion and Conclusions

We have investigated the predictability of the Moscow Stock Exchange. We are interested in a measure of market inefficiency that is not related to known sources of regularity in financial time series. Usually, these sources are not filtered out, and accordingly, their impact is taken into account in the degree of price predictability (see, e.g., [16,17,18]).

We have focused on two sources of regularity, namely volatility clustering and price staleness [20]. The process of filtering volatility clustering was performed in [19] by estimating volatility using the exponentially weighted moving average. We have developed a modification of the volatility estimation by taking into consideration the effect of price staleness. Price staleness produces excess 0-returns that affect the estimation of volatility. Another approach of estimating volatility in the case of presence 0-returns was proposed in [21] where all 0-returns are reevaluated during an expectation-maximization algorithm. In our approach, we separate 0-returns that may have resulted from rounding and from price staleness. Thus, we also filter out apparent inefficiency due to price staleness. Our approach combining the estimates of volatility and the degree of staleness can be used for a real-time causal analysis, since only past observations are used.

One of the clear advantage of the proposed approach relies in its simplicity: There is only one smoothing parameter in the method that can be optimized using historical data. We fix the value of smoothing parameter equaled 0.05. In the literature, the smoothing parameter α is usually taken close to 0. Using the principle of the best one-step forecasting, the smoothing parameter is set to 0.06 for the daily data and to 0.03 for the monthly data [33]. The value of the parameter α is set to be equal 0.12 for in-sample testing and 0.22 for out-of-sample testing in [46]. Hunter [32] suggests using α=0.2±0.1.

We used the Shannon entropy as a measure of randomness to infer the degree of (in)efficiency of the Moscow Stock Exchange. We used two types of the discretization of return time series to test efficiency more reliably for each month. The 4-symbol discretization helps find more price movements that lead to market inefficiency than compared to the 3-symbol discretization. Eighty percent of months over the period from 2012 to 2021 are defined as inefficient. According to Risso [24], a higher level of efficiency corresponds to more developed markets. Deviation from efficiency is a frequent phenomenon in various markets. For example, the authors of work [14] conclude that the Colombo Stock Exchange is only 10.5% efficient while the Pakistan Stock Exchange is 23.7% efficient. Cajueiro and Tabak [10] have shown that Asian markets are less efficient than Latin American markets. The authors of [28] estimates the efficiency of the Tunisian stock market as 97%. There are periods of inefficiency for some stocks traded at the Tel-Aviv stock exchange as stated in [15]. Short periods of inefficiency were also detected for US stock markets in [25].

By investigating the discretized values of filtered price returns, we came to the following conclusions:Even after filtering out all known sources of regularity, most months contain signals of market inefficiency.The most inefficient months are grouped together for two stocks exhibiting the lowest efficiency rates.For such months, discretized price returns before and after filtering out apparent inefficiencies are predictable.We introduced the entropy of co-movement. Stock prices display common patterns that have an interpretation in terms of the sector that the stock belong to.The stocks of banks and oil companies cluster together in terms of co-inefficiency for the case of the Moscow stock exchange.

One possible improvement to stock clustering is to modify the entropy of co-movement such that it is possible to define a proper distance function. This is left for future research. The proposed method for measuring market efficiency using the Shannon entropy can be applied in other markets of different countries. In this study, we used monthly time intervals for entropy calculation. Our future work will be related to the optimization of the length of return time series. One problem is to find a significant decrease in entropy without using Monte Carlo simulations. We also plan to switch to a higher frequency (less than one minute) to analyze the predictability of financial time series.

## Figures and Tables

**Figure 1 entropy-24-01184-f001:**
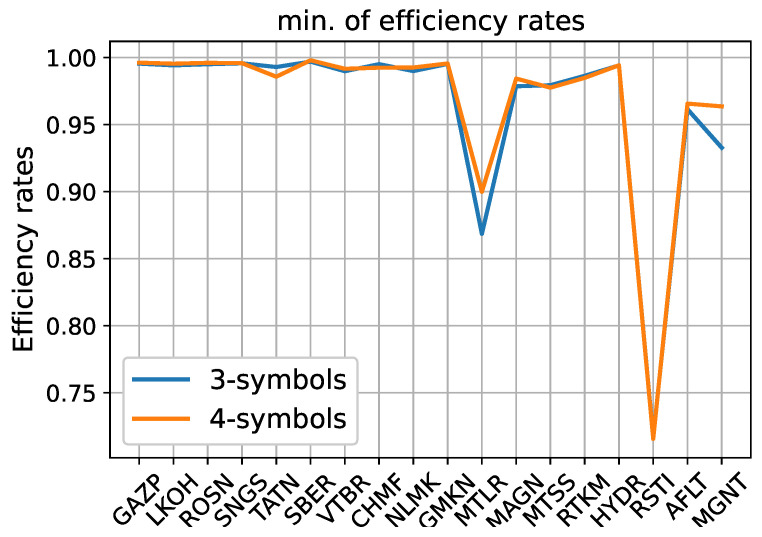
Minimum of efficiency rates for 18 stocks using 3- and 4-symbol discretizations.

**Figure 2 entropy-24-01184-f002:**
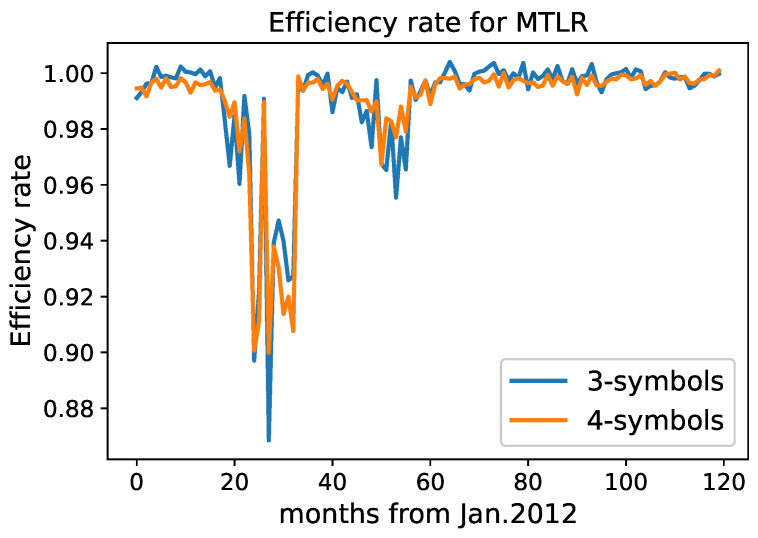
Efficiency rate for the MLTR stock using 3- and 4-symbol discretizations.

**Figure 3 entropy-24-01184-f003:**
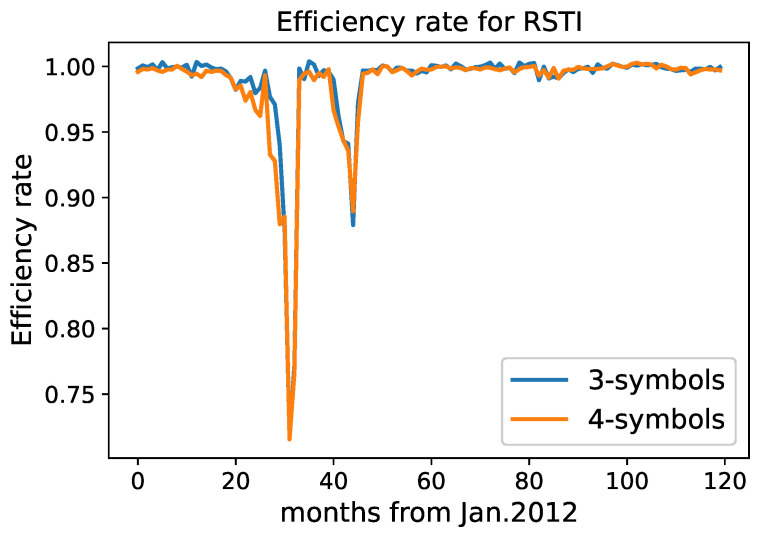
Efficiency rate for the RSTI stock using 3- and 4-symbol discretizations.

**Figure 4 entropy-24-01184-f004:**
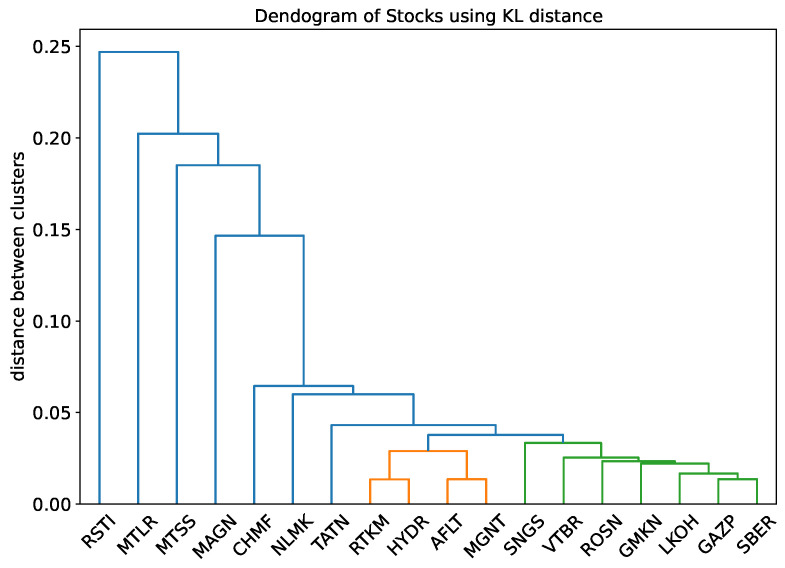
Hierarchical clustering tree using KL distance. The threshold for clustering into groups is 0.035.

**Figure 5 entropy-24-01184-f005:**
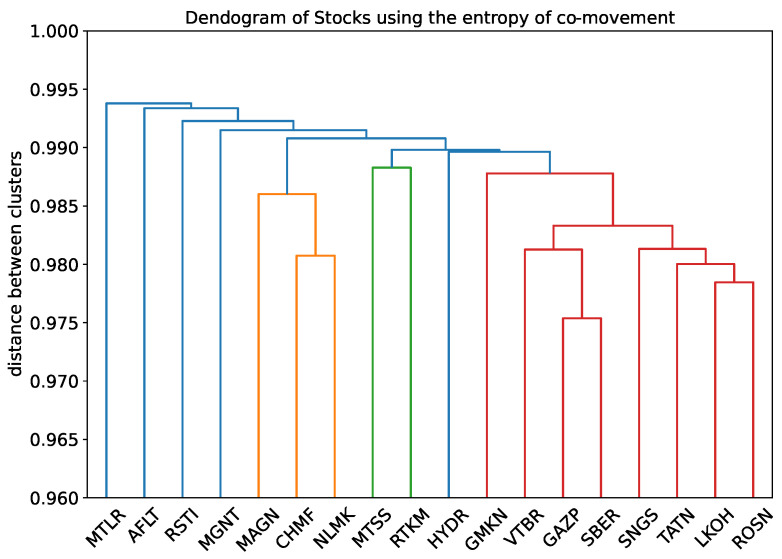
Hierarchical clustering tree using the entropy of co-movement. The threshold for clustering into groups is 0.989.

**Table 1 entropy-24-01184-t001:** Stocks of Russian companies traded at Moscow Exchange.

Ticker	Company	Sector	Size	Outliers
GAZP	Gazprom	Oil	1,307,427	50
LKOH	Lukoil	Oil	1,287,582	192
ROSN	Rosneft	Oil	1,270,592	130
SNGS	Surgutneftegaz	Oil	1,211,809	11
TATN	Tatneft	Oil	1,191,390	174
SBER	Sberbank	Bank	1,309,402	37
VTBR	VTB Bank	Bank	1,287,330	0
CHMF	Severstal	Metal	1,214,735	157
NLMK	Novolipetsk Steel	Metal	1,194,324	58
GMKN	Nornikel	Metal	1,272,769	197
MTLR	Mechel	Metal	1,084,990	161
MAGN	Magnitogorsk Iron and Steel Works	Metal	1,106,771	13
MTSS	Mobile TeleSystems	Telecommunications	1,153,527	260
RTKM	Rostelecom	Telecommunications	1,140,798	134
HYDR	RusHydro	Electric utility	1,252,584	0
RSTI	Rosseti	Electricity	1,094,244	0
AFLT	Aeroflot	Airline	1,083,552	123
MGNT	Magnit	Food retailer	1,184,223	544

For each company, we specify the ticker of stock, its sector, the size of data, and the amount of outliers removed. The size is given in the amount of minutes with trading activity.

**Table 2 entropy-24-01184-t002:** Results on volatility estimation.

Model	MAPE, Method $v_{1}$	MAPE, $v_{2}$	MAPE with α=0.05, $v_{1}$	MAPE with α=0.05, $v_{2}$	MAPE w/o 0-Filtering, $v_{1}$	MAPE w/o 0-Filtering, $v_{2}$
$\sigma_{1},q^{1}$	$\underset{\left(0.0007,0.0507\right)}{0.0193}$	$\underset{\left(0.0014,0.0406\right)}{0.017}$	$\underset{\left(0.0955,0.0995\right)}{0.0975}$	$\underset{\left(0.0878,0.0915\right)}{0.0897}$	$\underset{\left(0.0007,0.0507\right)}{0.0193}$	$\underset{\left(0.0014,0.0406\right)}{0.017}$
$\sigma_{1},q^{2}$	$\underset{\left(0.0245,0.1017\right)}{0.0607}$	$\underset{\left(0.0293,0.1057\right)}{0.0629}$	$\underset{\left(0.093,0.0972\right)}{0.095}$	$\underset{\left(0.0893,0.0936\right)}{0.0914}$	$\underset{\left(0.0459,0.131\right)}{0.0862}$	$\underset{\left(0.0294,0.1154\right)}{0.0674}$
$\sigma_{1},q^{3}$	$\underset{\left(0.0333,0.1278\right)}{0.0737}$	$\underset{\left(0.033,0.1338\right)}{0.0756}$	$\underset{\left(0.0928,0.0971\right)}{0.0948}$	$\underset{\left(0.0894,0.094\right)}{0.0915}$	$\underset{\left(0.0917,0.1863\right)}{0.138}$	$\underset{\left(0.0368,0.1592\right)}{0.0888}$
$\sigma_{1},q^{4}$	$\underset{\left(0.0323,0.1213\right)}{0.0716}$	$\underset{\left(0.0354,0.1268\right)}{0.0739}$	$\underset{\left(0.0926,0.0973\right)}{0.0949}$	$\underset{\left(0.089,0.0937\right)}{0.0913}$	$\underset{\left(0.1022,0.1875\right)}{0.1404}$	$\underset{\left(0.0405,0.1516\right)}{0.0873}$
$\sigma_{2},q^{1}$	$\underset{\left(0.1082,0.121\right)}{0.1121}$	$\underset{\left(0.1146,0.1244\right)}{0.1183}$	$\underset{\left(0.1438,0.1481\right)}{0.1459}$	$\underset{\left(0.1422,0.147\right)}{0.1446}$	$\underset{\left(0.108,0.1207\right)}{0.1118}$	$\underset{\left(0.1144,0.1243\right)}{0.1179}$
$\sigma_{2},q^{2}$	$\underset{\left(0.1163,0.1715\right)}{0.1359}$	$\underset{\left(0.1237,0.1765\right)}{0.1411}$	$\underset{\left(0.1439,0.1487\right)}{0.1462}$	$\underset{\left(0.1457,0.1526\right)}{0.1489}$	$\underset{\left(0.1043,0.1819\right)}{0.1341}$	$\underset{\left(0.1193,0.1832\right)}{0.1407}$
$\sigma_{2},q^{3}$	$\underset{\left(0.1198,0.1958\right)}{0.146}$	$\underset{\left(0.1266,0.1981\right)}{0.1519}$	$\underset{\left(0.1449,0.1499\right)}{0.1473}$	$\underset{\left(0.1464,0.1534\right)}{0.1496}$	$\underset{\left(0.1196,0.2271\right)}{0.1649}$	$\underset{\left(0.123,0.222\right)}{0.1589}$
$\sigma_{2},q^{4}$	$\underset{\left(0.1205,0.1912\right)}{0.146}$	$\underset{\left(0.1256,0.1986\right)}{0.15}$	$\underset{\left(0.1447,0.1498\right)}{0.1472}$	$\underset{\left(0.1463,0.1532\right)}{0.1494}$	$\underset{\left(0.1274,0.2261\right)}{0.1696}$	$\underset{\left(0.1223,0.2239\right)}{0.1571}$
$\sigma_{3},q^{1}$	$\underset{\left(0.1446,0.1513\right)}{0.1479}$	$\underset{\left(0.1442,0.1505\right)}{0.1473}$	$\underset{\left(0.1467,0.1522\right)}{0.1495}$	$\underset{\left(0.1442,0.1502\right)}{0.1473}$	$\underset{\left(0.1446,0.1513\right)}{0.1479}$	$\underset{\left(0.1441,0.1503\right)}{0.1472}$
$\sigma_{3},q^{2}$	$\underset{\left(0.1485,0.1891\right)}{0.1592}$	$\underset{\left(0.1508,0.1857\right)}{0.1613}$	$\underset{\left(0.149,0.1574\right)}{0.1529}$	$\underset{\left(0.1497,0.1598\right)}{0.1546}$	$\underset{\left(0.144,0.2033\right)}{0.1622}$	$\underset{\left(0.1491,0.1978\right)}{0.1628}$
$\sigma_{3},q^{3}$	$\underset{\left(0.1528,0.2048\right)}{0.1681}$	$\underset{\left(0.1556,0.2178\right)}{0.171}$	$\underset{\left(0.1525,0.1616\right)}{0.1567}$	$\underset{\left(0.1536,0.1639\right)}{0.1584}$	$\underset{\left(0.154,0.2477\right)}{0.1904}$	$\underset{\left(0.1546,0.2464\right)}{0.1815}$
$\sigma_{3},q^{4}$	$\underset{\left(0.1527,0.1997\right)}{0.1668}$	$\underset{\left(0.1555,0.2178\right)}{0.1701}$	$\underset{\left(0.1525,0.1613\right)}{0.1568}$	$\underset{\left(0.1537,0.1633\right)}{0.1583}$	$\underset{\left(0.1591,0.246\right)}{0.192}$	$\underset{\left(0.1556,0.2455\right)}{0.181}$
$\sigma_{4},q^{1}$	$\underset{\left(0.1856,0.1952\right)}{0.1897}$	$\underset{\left(0.1838,0.1911\right)}{0.1873}$	$\underset{\left(0.1844,0.1918\right)}{0.1881}$	$\underset{\left(0.1879,0.1968\right)}{0.1924}$	$\underset{\left(0.1856,0.1952\right)}{0.1897}$	$\underset{\left(0.1837,0.1911\right)}{0.1873}$
$\sigma_{4},q^{2}$	$\underset{\left(0.1906,0.2454\right)}{0.2035}$	$\underset{\left(0.1921,0.2474\right)}{0.2057}$	$\underset{\left(0.1891,0.2022\right)}{0.1954}$	$\underset{\left(0.1961,0.2119\right)}{0.2037}$	$\underset{\left(0.1836,0.2617\right)}{0.2049}$	$\underset{\left(0.1902,0.2642\right)}{0.2079}$
$\sigma_{4},q^{3}$	$\underset{\left(0.1965,0.2623\right)}{0.2146}$	$\underset{\left(0.1996,0.2757\right)}{0.2166}$	$\underset{\left(0.1951,0.2077\right)}{0.2015}$	$\underset{\left(0.2026,0.2177\right)}{0.2101}$	$\underset{\left(0.1912,0.307\right)}{0.2318}$	$\underset{\left(0.1988,0.3082\right)}{0.2294}$
$\sigma_{4},q^{4}$	$\underset{\left(0.1967,0.2591\right)}{0.214}$	$\underset{\left(0.1986,0.2689\right)}{0.2155}$	$\underset{\left(0.1951,0.2088\right)}{0.2013}$	$\underset{\left(0.2023,0.2185\right)}{0.2097}$	$\underset{\left(0.1976,0.3064\right)}{0.2338}$	$\underset{\left(0.1988,0.306\right)}{0.2286}$

The first column indicated a model. Columns 2 and 3 represent results for two methods described in Section 2.2.3. Columns 4 and 5 are for the same methods but with the fixed value of α. Columns 6 and 7 shows the error of
the standard EMWA approach with the optimal selected value of α. Values highlighted in bold are the smallest errors in each row. The 95% CI is presented below each averaged statistic. $v_{1}$ stands for using $Sig_{1}$; $v_{2}$ stands for using $Sig_{2}$.

**Table 3 entropy-24-01184-t003:** Results upon filtering out 0-returns.

Model	α for $v_{1}$	α for $v_{2}$	${\mathrm{Er}}_{N}$, $v_{1}$	${\mathrm{Er}}_{N}$, $v_{2}$	Fraction of Data Deleted, $v_{1}$	Fraction of Data Deleted, $v_{2}$
$\sigma_{1},q^{1}$	$\underset{\left(0.0,0.0137\right)}{0.0027}$	$\underset{\left(0.0,0.0103\right)}{0.0022}$	$\underset{\left(0.0,0.0\right)}{0.0006}$	$\underset{\left(0.0,0.0259\right)}{0.0015}$	$\underset{\left(0.0,0.0\right)}{0.0001}$	$\underset{\left(0.0,0.0052\right)}{0.0003}$
$\sigma_{1},q^{2}$	$\underset{\left(0.0033,0.0569\right)}{0.0228}$	$\underset{\left(0.0044,0.067\right)}{0.0259}$	$\underset{\left(0.0004,0.026\right)}{0.0094}$	$\underset{\left(0.0005,0.0295\right)}{0.011}$	$\underset{\left(0.0814,0.3244\right)}{0.2005}$	$\underset{\left(0.0818,0.3247\right)}{0.2008}$
$\sigma_{1},q^{3}$	$\underset{\left(0.006,0.0902\right)}{0.0335}$	$\underset{\left(0.0058,0.1063\right)}{0.0379}$	$\underset{\left(0.0005,0.0288\right)}{0.0106}$	$\underset{\left(0.0005,0.0336\right)}{0.0121}$	$\underset{\left(0.2474,0.481\right)}{0.3661}$	$\underset{\left(0.246,0.4797\right)}{0.3659}$
$\sigma_{1},q^{4}$	$\underset{\left(0.0056,0.0824\right)}{0.0314}$	$\underset{\left(0.007,0.0966\right)}{0.036}$	$\underset{\left(0.0004,0.0283\right)}{0.0104}$	$\underset{\left(0.0007,0.0355\right)}{0.0122}$	$\underset{\left(0.2521,0.4717\right)}{0.3628}$	$\underset{\left(0.2515,0.4713\right)}{0.3626}$
$\sigma_{2},q^{1}$	$\underset{\left(0.0,0.0161\right)}{0.0039}$	$\underset{\left(0.0006,0.0146\right)}{0.0037}$	$\underset{\left(0.0,0.0438\right)}{0.0149}$	$\underset{\left(0.0,0.0586\right)}{0.035}$	$\underset{\left(0.0,0.0092\right)}{0.0029}$	$\underset{\left(0.0,0.0134\right)}{0.0067}$
$\sigma_{2},q^{2}$	$\underset{\left(0.0059,0.0903\right)}{0.035}$	$\underset{\left(0.0054,0.1021\right)}{0.0367}$	$\underset{\left(0.0012,0.0448\right)}{0.0209}$	$\underset{\left(0.0039,0.0601\right)}{0.0319}$	$\underset{\left(0.0856,0.3249\right)}{0.2016}$	$\underset{\left(0.0878,0.3263\right)}{0.2032}$
$\sigma_{2},q^{3}$	$\underset{\left(0.0079,0.1326\right)}{0.0489}$	$\underset{\left(0.0091,0.1476\right)}{0.0556}$	$\underset{\left(0.001,0.0473\right)}{0.0217}$	$\underset{\left(0.0022,0.0603\right)}{0.0275}$	$\underset{\left(0.25,0.4835\right)}{0.3706}$	$\underset{\left(0.2518,0.4836\right)}{0.371}$
$\sigma_{2},q^{4}$	$\underset{\left(0.0093,0.1268\right)}{0.049}$	$\underset{\left(0.0082,0.1484\right)}{0.0525}$	$\underset{\left(0.0012,0.0443\right)}{0.0206}$	$\underset{\left(0.0013,0.0571\right)}{0.0274}$	$\underset{\left(0.2495,0.4695\right)}{0.3645}$	$\underset{\left(0.2491,0.4692\right)}{0.3651}$
$\sigma_{3},q^{1}$	$\underset{\left(0.0349,0.0527\right)}{0.0424}$	$\underset{\left(0.0392,0.0603\right)}{0.048}$	$\underset{\left(0.0,0.0337\right)}{0.0034}$	$\underset{\left(0.0,0.0402\right)}{0.0089}$	$\underset{\left(0.0,0.0067\right)}{0.0007}$	$\underset{\left(0.0,0.0085\right)}{0.0018}$
$\sigma_{3},q^{2}$	$\underset{\left(0.0267,0.1456\right)}{0.0672}$	$\underset{\left(0.0249,0.1592\right)}{0.0767}$	$\underset{\left(0.0006,0.0404\right)}{0.0155}$	$\underset{\left(0.0009,0.0551\right)}{0.02}$	$\underset{\left(0.0826,0.3248\right)}{0.1995}$	$\underset{\left(0.0826,0.3251\right)}{0.1997}$
$\sigma_{3},q^{3}$	$\underset{\left(0.0264,0.1734\right)}{0.0825}$	$\underset{\left(0.0301,0.2371\right)}{0.0985}$	$\underset{\left(0.0011,0.0477\right)}{0.018}$	$\underset{\left(0.0008,0.0702\right)}{0.0243}$	$\underset{\left(0.2444,0.4746\right)}{0.3678}$	$\underset{\left(0.2421,0.4751\right)}{0.3671}$
$\sigma_{3},q^{4}$	$\underset{\left(0.0266,0.163\right)}{0.0788}$	$\underset{\left(0.0325,0.2362\right)}{0.0969}$	$\underset{\left(0.001,0.0463\right)}{0.0178}$	$\underset{\left(0.0007,0.067\right)}{0.0222}$	$\underset{\left(0.2466,0.476\right)}{0.3623}$	$\underset{\left(0.2486,0.4739\right)}{0.3615}$
$\sigma_{4},q^{1}$	$\underset{\left(0.0696,0.1037\right)}{0.0819}$	$\underset{\left(0.0757,0.119\right)}{0.0904}$	$\underset{\left(0.0,0.0248\right)}{0.0013}$	$\underset{\left(0.0,0.0329\right)}{0.0047}$	$\underset{\left(0.0,0.0052\right)}{0.0003}$	$\underset{\left(0.0,0.0075\right)}{0.0011}$
$\sigma_{4},q^{2}$	$\underset{\left(0.0534,0.2359\right)}{0.1132}$	$\underset{\left(0.0576,0.2925\right)}{0.1339}$	$\underset{\left(0.0007,0.0564\right)}{0.0185}$	$\underset{\left(0.0008,0.087\right)}{0.0265}$	$\underset{\left(0.0765,0.3287\right)}{0.1993}$	$\underset{\left(0.077,0.3257\right)}{0.1982}$
$\sigma_{4},q^{3}$	$\underset{\left(0.0557,0.2678\right)}{0.1338}$	$\underset{\left(0.0597,0.3734\right)}{0.1596}$	$\underset{\left(0.0009,0.0621\right)}{0.0214}$	$\underset{\left(0.001,0.1119\right)}{0.0321}$	$\underset{\left(0.2419,0.4823\right)}{0.3687}$	$\underset{\left(0.2378,0.4817\right)}{0.3669}$
$\sigma_{4},q^{4}$	$\underset{\left(0.0571,0.263\right)}{0.1317}$	$\underset{\left(0.0613,0.3541\right)}{0.1556}$	$\underset{\left(0.0008,0.0667\right)}{0.0211}$	$\underset{\left(0.0011,0.108\right)}{0.0315}$	$\underset{\left(0.2599,0.4823\right)}{0.3641}$	$\underset{\left(0.2564,0.4822\right)}{0.3625}$

Values of α, errors of the number of 0-returns due to rounding, and fraction of data set as missing values. The first column indicated a model. 95% CI is presented below each averaged statistic. $v_{1}$ stands for using $Sig_{1}$; $v_{2}$ stands for using $Sig_{2}$.

**Table 4 entropy-24-01184-t004:** The degree of inefficiency for each stock.

Ticker	Degree of Inefficiency	For 3 Symbols Only	For 4 Symbols Only
GAZP	0.725	0.392	0.675
LKOH	0.65	0.342	0.542
ROSN	0.742	0.392	0.708
SNGS	0.725	0.4	0.625
TATN	0.617	0.392	0.525
SBER	0.725	0.433	0.658
VTBR	0.842	0.592	0.792
CHMF	0.858	0.55	0.692
NLMK	0.8	0.467	0.692
GMKN	0.733	0.475	0.608
MTLR	0.992	0.783	0.975
MAGN	0.833	0.65	0.758
MTSS	0.967	0.7	0.942
RTKM	0.942	0.683	0.908
HYDR	0.892	0.75	0.8
RSTI	0.917	0.742	0.875
AFLT	0.983	0.775	0.95
MGNT	0.842	0.667	0.742

Fraction of inefficient months using 3-symbol and 4-symbol discretizations. Each value in the last two columns is calculated using 120 efficiency rates.

**Table 5 entropy-24-01184-t005:** The most frequent blocks appearing for the MLTR stock and the probabilities of success.

Months of 2014	The Most Frequent Block, 3-s	The Most Frequent Block, 4-s	Prob. of Success, Filtered	Prob. of Success, Original
January	00000	1111	0.64	0.75
February	00000	2222	0.64	0.74
May	00000	1111	0.61	0.73
June	22222	1111	0.60	0.73
July	11111	1111	0.62	0.74
August	00000	1111	0.61	0.76
September	00000	1111	0.63	0.74
October	120120	0303	0.55	0.6

The first column represents months with the lowest efficiency rates. Columns 2 and 3 are the most frequent blocks in 3- and 4-symbol discretizations. Columns 4 and 5 include the probability of success when applying the simple trading strategy for filtered and original price returns.

**Table 6 entropy-24-01184-t006:** The most frequent blocks appearing for the RSTI stock and probabilities of success.

Months	The Most Frequent Block, 3-s	The Most Frequent Block, 4-s	Prob. of Success, Filtered	Prob. of Success, Original
April 2014	212121	0111	0.63	0.77
May 2014	00000	1111	0.61	0.73
June 2014	00000	1111	0.6	0.73
July 2014	00000	2222	0.62	0.74
August 2014	00000	2222	0.61	0.76
September 2014	000000	22222	0.63	0.74
June 2015	00000	2222	0.54	0.61
July 2015	00000	1111	0.55	0.6
August 2015	00000	2222	0.54	0.6
September 2015	00000	2222	0.55	0.61
October 2015	11111	0111	0.56	0.62

The first column represents months with the lowest efficiency rates. Columns 2 and 3 are the most frequent blocks in 3- and 4-symbol discretization. The length of a block is defined using Equation (Equation 5). Columns 4 and 5 are the probability of the success of the simple trading strategy for filtered and original price returns.

## Data Availability

The Python codes (v. 3.9.5) used in this study are available in Appendix L. The dataset analyzed in this study can be found here: https://www.finam.ru/profile/moex-akcii/gazprom/export/ (accessed on 17 August 2022).

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
