# Peer review of "Efficiency of the Moscow Stock Exchange before 2022"

_entropy, 2022, doi:10.3390/e24091184_

Round 1

Reviewer 1 Report

In general I have enjoyed reading this manuscript. I have no major issues with the paper, but have quite a few minor comments, which I list below.

I would think that literature review would benefit from comparison of the approach by the authors against one taken when analyzing Bitcoin data [https://doi.org/10.1063/1.5036517]. Also I think that introduction would benefit from discussion on another approach to market efficiency - (multi)fractal perspective, e.g., as in [https://doi.org/10.1016/j.physa.2016.12.037].

Eq. (1) is not sufficiently explained. Some assumptions appear to be hidden. For example, selected value for \mu_1 appears to imply that \r_i are normally distributed, while high-frequency returns are known to have fatter tails, which are better fitted by a power law than by exponential.

It is not clear whether Eq. (1) or Eq. (2) is actually used in the article. Likely optimal value of \alpha would also depend on which of the equations is actually used.

Line 123, brackets in the mathematical expression of the minimized value are missing. Likewise on the line 124.

Line 146-148. Slightly confusing as one might interpret that Eq. (3) is the probability that 0-return value is kept. In fact the probability should be p_i divided by the empirical frequency of 0-returns. Is that correct?

Line 161-163. Likely I am missing something (related to the previous comment). Though, how is it possible to process data in real time if one needs to know empirical frequency of 0-returns?

Eq. (4) I would believe that it would be easier to read if proper spacing is added to this expression and the previous one. Also & character could be replace with proper mathematical notation (union?) or simply word "and".

Section 2.7, 3-symbols definition. Why lowest tertile is encode as 1? It would be easier to read if the encoding of partitions would be sequential.

Eq. (5) missing closing parenthesis. The meaning of this equation is unclear. Why such condition is imposed on k?

Line 201. K-L divergence measures distance between two distribution. Not necessarily between two sequences, as sequences also include temporal information (e.g., correlation).

Section 3.1. "pr_i" looks kind of strange. It would be better to use single symbol, not a pair of symbols, as notation for a single things.

Section 3.1. Top equation of page 8, shouldn't the last pr_t be pr^4_t? It is unclear what the numbers mean in the ARCH and GARCH parentheses. Typically it is either order of the process or the parameter values of the process. But in the later case an explanation should be added. Also some discussion on why such numbers are considered would be worthwhile.

Line 233 and 6th footnote. Could efficiency of the market depend on the timescale chosen for analysis? For example, market on longer time scales incorporating information just fine, but failing to catch up on the highest frequencies? Similarly to what is happening in agent-based model from [https://doi.org/10.1016/j.physa.2019.03.059].

Line 233 and 6th footnote. Could you meaningfully compare results for US ETF market and Moscow Stock Exchange? Or the amount MSE data is insufficient / US ETF market data not detailed enough?

Tables 2 and 3. "pr^1" has become "pr_1"?

Table 4. How overall degree of inefficiency is obtained?

Line 261. "parts" -> "groups"? And further in the text. "parts" might be understood as partition within the blocks themselves.

Lines 263-265. Sentences unclear. It seems predicates are missing?

Table 5 and 6. Why column 2 contains more than 3 digits?

Author Response

We thank the reviewer for careful reading of the paper and helpful comments. Please see the attachment.

Reviewer 2 Report

The subject of the article is interesting and worth describing. However, the method of implementation requires correction. The authors in the Introduction presented an introduction to the subject. The Introduction section is deficient. It does not contain all the necessary elements. The main goal and specific goals were not clearly defined. The Introduction section should contain research hypotheses or research questions.

The layout of the work is not entirely correct. I have already listed the items that can be found in section 1 Introduction. In section 2, individual subsections can be created. This is also what the authors do, but I have doubts about their large number. Section 2 should provide more detailed information on the scope of the research and outline the stages of the research. A flowchart may be prepared to help readers understand the stages of the research.

The title of section 4 should be different. Discussion and Conclusions should be separated into separate sections. A separate Discussion section is missing from this article. I understand a discussion as referring to other studies after presenting my research results. In my opinion, doing research without a clear comparison and reference to other research results in the fact that the obtained results cannot be properly assessed. Such references have appeared in other parts. Some are in section 2 and some in section 4. You can move some text to the Discussion section. The discussion should be broadened and the results achieved rather than methods should be referred to.

Conclusion section is incomplete. You must certainly refer to the hypotheses or research questions posed. Is it possible to verify the hypotheses positively or negatively? The conclusions can be bulleted. This section is too long. Conclusions should be a synthesis.

Other remarks:

Table 1. Size is in the penultimate column. In which unit? Is that the number of shares? or the number of minutes?

Line 88. The authors write about "methodology". It's an error. Methodology is the study of scientific research methods, their effectiveness, and their cognitive value. Certainly, the author does not want to study research methods.

Figure 4. There is no title for the Y axis. The same applies to Figure 5.

Author Response

We thank the reviewer for the useful comments about the structure of the paper. Please see the attachment.
